# Full-Face Snorkeling Masks Carry a Risk of Hypercapnia and Drowning in Younger Children: A Case Series

**DOI:** 10.3390/children12091148

**Published:** 2025-08-29

**Authors:** Laura Trapani, Federico Poropat, Elisabetta Cattaruzzi, Egidio Barbi, Chiara Zanchi

**Affiliations:** 1Clinical Department of Medical, Surgical and Health Science, University of Trieste, 34127, Trieste, Italy; egidio.barbi@burlo.trieste.it; 2Institute for Maternal and Child Health IRCCS Burlo Garofolo, 34137 Trieste, Italy; federico.poropat@burlo.trieste.it (F.P.); elisabetta.cattaruzzi@burlo.trieste.it (E.C.); chiara.zanchi@burlo.trieste.it (C.Z.)

**Keywords:** full-face snorkeling marks, drowning, children, respiratory arrest, hypercapnic hypoxemia

## Abstract

**Background**: Recently, a new type of full-face snorkeling mask (FFSM), called “Easy-breath” masks, has become extremely popular both in adults and children due to their effective marketing and relative comfort. However, these masks are complex engineering systems that, in case of malfunctioning or if used by young children, may readily cause CO_2_ rebreathing, especially in young children. **Case Presentation**: We present three cases of children under six years of age admitted to the emergency department, with two of them due to non-fatal drowning incidents and one following a cardiac arrest induced by drowning. All incidents occurred during brief submersions while using full-face snorkeling masks. **Conclusions**: When inappropriately used by younger children, full-face snorkeling masks may have a mechanical dead space larger than tidal volume, with a significant increased risk of rebreathing of CO_2_ and consequent risk for hypercapnic hypoxia. The hypercapnia may cause dizziness and respiratory distress, while hypoxia may cause confusion. Both may lead to loss of consciousness, which could be a potential cause of drowning, particularly in younger children.

## 1. Introduction

Drowning is a serious public health concern, whose highest morbidity and mortality are reported among children [1,2]. In recent years, a new type of full-face snorkeling mask (FFSM), called “Easy-breath” masks, has become popular, promising a 180-degree panoramic underwater vision and the ability to breathe naturally through both the nose and mouth, eliminating the discomfort of a traditional snorkel. While models suitable for children exist, manufacturers do not recommend their use under six years of age [3].

However, the apparent simple design of this mask hides significant engineering complexity and inherent risks that are not immediately apparent to the consumer. Indeed, various unidirectional valves are employed to ensure proper airflow to the user, while separate valves are responsible for expelling exhaled air from the device, so that, in the event of a device malfunctioning or a component deterioration, the user may be at risk of CO_2_ rebreathing.

In the current literature, there are concerns about the potential for rebreathing exhaled gas high in carbon dioxide. Reports of accidents, including fatalities, associated with the use of FFSMs [4], particularly in tourist destinations like Hawaii [5], have raised significant alarm in the aquatics and medical safety communities.

These fatalities have been reported to the nonprofit agency Divers Alert Network (DAN) for snorkelers who were using FFSMs. For these reasons, organizations such as Duke University have initiated specific studies in collaboration with academic research institutions to investigate these risks.

Farrel et al. [6] conducted a study analyzing the safety of these devices, documenting increased breathing resistance in FFSMs caused by water intrusion, which may lead to elevated respiratory distress among snorkelers during real-world use.

In this report, we discuss three cases of children under six years of age admitted to the emergency department of the Institute for Maternal and Child Health Burlo Garofolo of Trieste. Two of these children were admitted due to non-fatal drowning incidents, whereas the third was admitted following a cardiac arrest induced by drowning. All incidents occurred in the last two summers, during brief submersions, supervised by parents, while using full-face snorkeling masks.

In our opinion, these three cases represent the clinical manifestation of predictable risks due to the physiological tidal volume of young children and the mechanical dead volume of air of the device.

Ethical committee approval was not required according to Italian law since General Authorization to Process Personal Data for Scientific Research Purposes (Authorization no. 9/2014) declared that retrospective archive studies that use ID codes, preventing the data from being traced back directly to the data subject, do not need ethics approval [7]).

Furthermore, according to the Research Institute policy, parents of admitted children are requested to sign an informed consent form for the treatment of anonymized data for research purposes.

## 2. Case Series

### 2.1. Case 1

A five-year-old girl was snorkeling with her full-face mask when she had difficulty swimming and was saved by a swimmer. When he pulled the child out of the water, she appeared confused and presented with labial cyanosis, difficulty breathing, and an episode of vomiting. The girl did not showcase any sign of agitation when the swimmer saved her.

At admission in the ED, physical examination was remarkable for tachycardia (130 times/minute) and tachypnea (40 times/minute); peripheral oxygen saturation was 92% with slight intercostal retractions and wheezing on lungs auscultation. Blood tests showed increased leucocytes (25,000/mmc, N 16,530/mmc) and sodium level (147 mEq/L), with pH 7.32, pCO_2_ 40 mmHg, and HCO_3_^−^ 20 mEq/L. A chest X-ray revealed a bilateral lung infiltration, with a consolidation area over the right lung (Figure 1). The patient received high-flow oxygen therapy for 24 h with a full recovery.

### 2.2. Case 2

A six-year-old boy was swimming with his full-face mask when he started feeling confused and agitated; a family member helped him remove the mask and took him to the emergency room for respiratory distress.

He reported that some water had entered the mask and that he had not been able to remove the device. At evaluation in the ED, he presented tachypnea (36 breaths/minute), dyspnea with nodding and indrawing, hypoxia (peripheral oxygen saturation was 82%), and respiratory acidosis (pH 7.29, pCO_2_ 44 mmHg, HCO_3_^−^ 20 mEq/L, and BE −4.9). A chest X-ray showed bilateral thickening of the lungs’ interstitial tissue (Figure 2). He was transferred to the intensive care unit, receiving high-flow oxygen therapy for 24 h. He fully recovered uneventfully in the following days.

### 2.3. Case 3

A four-year-old girl was swimming in the pool with her mother, wearing her brother’s full-face mask. Suddenly, the child’s movements progressively slowed down until she became completely hypotonic. Someone subsequently took the child out of the water, and she was unresponsive and unconscious, with the face mask filled with water. A pediatrician happened to be at the scene and began CPR, which lasted for five cycles with referred ROSC.

At the end of the resuscitation maneuvers, the child was conscious, breathing spontaneously. She was transferred to the local hospital where she seemed to be lethargic and poorly responsive to painful stimuli; bilateral crackles were reported at chest auscultation. She was subsequently airlifted to our emergency department.

At admission, four hours after the initial event, the patient was alert, responsive, and oriented to time and place. Her vitals indicated sinus tachycardia (145 bpm), tachypnea (34 breaths/minutes), and chest auscultation revealed some wet crackles; SaO_2_ was 98%. Lung ultrasound showed few scattered non-confluent B-lines, suggesting the presence of a mild wet lung. Given the stable clinical condition, normal peripheral oxygen saturation, and balanced capillary blood gas analysis (pH 7.37, pCO_2_ 38.7 mmHg, and HCO_3_^−^ 22 mEq/L), a watchful waiting approach was preferred. Blood tests detected a mild elevation of liver enzymes (alanine aminotransferase = 208 U/L) and of troponin (25 ng/L). Her general and clinical conditions improved gradually over time.

No additional therapeutic interventions were required; therefore, the child was monitored clinically at close intervals and discharged after 24 h of observation.

## 3. Discussion

We report the cases of three children who could have developed severe and potentially fatal complications from the use of “Easy-breath” full-face masks, which are becoming increasingly widespread worldwide. The pathophysiological risks of using FFSMs in younger children are due to the risk of CO_2_ rebreathing and to the consequent risk of developing hypercapnic hypoxia.

To fully understand the risks of the use of full-face snorkeling masks in younger children, the functioning of full-face snorkeling masks and the mechanical dead spaces of underwater breathing devices need to be well understood.

The full-face snorkeling mask design is based on unidirectional airflow, which is achieved through internal compartmentalization. Fresh air is drawn in through the snorkel tube at the top of the mask. From there, it is supposed to flow through the main viewing chamber (eye zone). This design serves the dual purpose of providing breathable air and preventing visor fogging.

Fresh air then enters the sealed buccal–nasal pocket through one-way non-return valves.

Exhaled air, enriched with carbon dioxide, is supposed to be expelled from the buccal–nasal area through a separate pathway, which is represented by lateral channels that run along the sides of the mask and exit at the top of the snorkel through dedicated exhalation valves (Figure 3). Theoretically, this configuration is intended to prevent the mixture of inhaled (fresh and rich in O_2_) and exhaled (rich in CO_2_) air.

This design’s critical point and principal vulnerability lie in the integrity of the seals and valves that separate the inhalation and exhalation pathways. The engineering risks associated with FFSMs emerged unexpectedly during the COVID-19 pandemic with a global effort to repurpose snorkeling masks as devices to deliver continuous positive airway pressure (CPAP) [8].

These studies identified several critical failure risks in snorkeling masks under controlled conditions: dangerous CO_2_ buildup, problems with the buccal–nasal seal, and malfunctions of the unidirectional valves [3]. The silicone seals, meant to separate the breathing pocket from the viewing chamber, may degrade over time or suffer from manufacturing flaws, reducing their effectiveness. Likewise, the unidirectional valves essential for proper airflow can fail by sticking open or closed, or by becoming obstructed with debris like sand or salt, all of which compromise safe breathing.

As a result, during exhalation, a portion of the exhaled, CO_2_-rich air may leak into the viewing chamber, contaminating fresh air and leading to carbon dioxide rebreathing and potentially hypercapnic hypoxemia [3]. Due to the presence of rebreathed air in the viewing chamber, visor fogging may represent the first warning sign of malfunctioning of the device.

It should be noted that even the traditional mask and snorkeler system may also pose a risk for CO_2_ rebreathing due to the possibly lower physiological tidal volume of a person compared to the mechanical dead volume of a device. In fact, if all the exhaled air is not fully expelled from the tube during an exhalation, the snorkeler will re-inhale CO_2_-rich air. A traditional snorkel has an internal volume (which corresponds to the mechanical dead space of the device) of about 160 mL [3]. When the full-face snorkeling mask is functioning as intended, the mechanical dead space of the nasal–buccal pocket is about 250 mL. Adding the snorkel volume itself, the total respiratory dead space (the volume that must be flushed with fresh air before it reaches the lungs) was measured between 250 mL and 610 mL [9]—an increase in dead space of 56% to nearly 300% compared to a traditional snorkel.

If the seals or valves are not working properly, this dead volume can increase to up to 1470 mL depending on the brand [9]. A larger “equipment dead space” results in greater risk of hypercapnic hypoxia.

The risks of rebreathing and its consequences when using FFSMs compared to a conventional snorkel had been studied in twenty healthy adults by Grundemann et al. [3]. In this study, twenty healthy participants aged 18 to 60 years were enrolled in a dry environment. They were asked to wear three types of snorkel equipment (two FFSM and one conventional snorkel) in three conditions: rest, light-intensity exercise, and moderate-intensity exercise on a cycle ergometer. They were continuously monitored, and numerous factors were constantly monitored and collected: peripheral oxygen saturation; respiratory rate; pCO_2_ and pO_2_ (both within the FFSM’s eye-pockets and at the oronasal compartment; instead, the conventional snorkel had a sampling line ported approximately 5 cm from the mouthpiece).

Light-intensity exercise tests that ran with the FFSMs had to be discontinued in 45% of the cases after exceeding 7.0 kPa end-tidal CO_2_ (considered a safety threshold) compared to conventional snorkels which were only discontinued in 20% of the cases. These results are replicable even in moderate-intensity exercise tests. The pCO_2_ and pO_2_ in the eye-pockets of the FFSMs fluctuated and were significantly higher (pCO_2_) and lower (pO_2_) compared to fresh air, which clearly indicated rebreathing in all FFSM wearers [3].

If these findings are valid for an adult population, in which the estimated tidal volume is approximately 500 mL, a significantly higher risk is posed in the pediatric population, where tidal volume is directly proportional to a child’s weight: the standard reference values for pediatric tidal volume (Vt) range between 7 and 10 mL/kg of body weight [10].

In applying these values to our cases, we can forecast that, in a 20 kg child, the estimated tidal volume may be 140–200 mL, and in smaller children this value will become alarmingly lower. Remarkably, a healthy, third-percentile six-year-old child will have a weight as low as 16 kg with a lower tidal volume.

Additionally, full-face masks are equipped with a specialized system that prevents water from entering the snorkel when submerged by waves or when the user tilts their head too far downwards into the water. This system consists of a float within the snorkel that rises to seal the opening if the person submerges too deeply. To clear the snorkel, the user must exhale forcefully to lift the float back above the water level. It is plausible that smaller children may lack the necessary strength to perform this action effectively.

The tighter-fitting head straps of the full-face design may make removing this device a complex operation in case of emergencies, much more complicated than simply spitting out a snorkel, leading to agitation and increased respiratory rate, paired with a tendency towards shallow breathing that causes a consistent reduction in fresh air reaching down towards the functioning alveoli. Conversely, in case the head straps are not perfectly adherent to the skin, water might get into the mask, leading to water aspiration risks.

It can be concluded that the use of the full-face masks may be dangerous for a variety of reasons (see Table 1): (1) the risk of rebreathing; (2) the potential failure of some safety mechanisms of the mask, such as valves and internal compartmentalization; (3) the mask’s design may make it difficult for children to remove the device in case of an emergency.

For these reasons, the use of these devices in pediatric population can have disastrous consequences, with a greater tendency to hypercapnia and hypoxia.

The hypercapnia may cause dizziness, respiratory distress, headaches, and loss of consciousness, while hypoxia may cause confusion and ultimately loss of consciousness. These effects can be both dangerous and even fatal while snorkeling, especially in children.

It must be highlighted that, in all the cases presented above, an adult was present at the scene, leading to the rescue of the child, and that two of them appeared motionless and drowsy without movements, suggesting agitation related to awareness of drowning. Hypercapnic hypoxia may in fact be subtle, and children in need may only slow down motion and faint without showing agitation.

In clinical practice, it is often difficult to measure CO_2_ levels and hypoxemia through blood gas analysis if the child quickly resumes breathing without a full-face snorkel mask. Since both hypercapnic and hypoxemic states are rapidly reversible, blood gas values may normalize within a short interval after the accident, as occurred in the third case, where the blood gas analysis was entirely normal.

Nonetheless, even shortly after an episode of hypercapnic hypoxemia, a mild acidosis may persist.

The acute rise in hematic PaCO_2_ causes a fall in pH, reflecting the consumption of bicarbonate as compensation for respiratory acidosis.

Renal retention of bicarbonate occurs subsequently but requires time to affect the normalization of the pH. For this reason, an altered pH may persist longer and serve as a more enduring indicator of a transient hypercapnic hypoxic episode, as observed in the first two cases.

## 4. Limits

Our study does not allow us to draw generalized conclusions about the incidence or prevalence of hypercapnic hypoxia related to FFSM use in children. Moreover, we did not investigate different kinds of mask, and different manufacturers may have products of different quality, carrying different risks. We could also not obtain detailed data about the dead space volume of the different pediatric masks. Furthermore, we did not specifically investigate how mask maintenance and cleaning, which may play a role in valve obstruction, were performed by parents. The purpose of this report is to describe sentinel events. The strength of this report lies in the consistency between clinical observation, established physiological principles, and available engineering data.

## 5. Conclusions

Drowning is a serious public health problem that particularly affects children. According to the WHO-UNICEF report, drowning is the second leading cause of death in children after road accidents. According to estimates, approximately 480 children worldwide die every day from drowning. For this reason, all preventive efforts should be considered to reduce this risk.

This report, along with the already available literature, reinforces concerns about the safety of FFSMs, especially in children under the age of six, and how FFSM use can pose a serious risk of hypercapnic hypoxia. A user has no way to visually inspect or test the integrity of internal seals and valves before use, which contributes to the potential unsafety of these masks. The difference between a safe mask and a dangerous one can be an invisible manufacturing defect or invisible material deterioration. Even if the idea of a pre-school child using such a swimming device appears unlikely, suggesting it as a rare event, these cases show that this situation is not uncommon.

While manufacturers’ instructions clearly advise against using such masks for children under six years of age and clearly specify that it is always mandatory for an adult to be present to assist any child in need, age alone is not a clear discriminator for safe mask usage. Conversely, weight is significantly correlated to tidal volume. Therefore, a more scientifically appropriate recommendation from manufacturers should involve a minimum weight (25–30 kg) rather than a minimum age—or both weight and age.

This study aims to raise awareness among pediatricians of the risks associated with FFSMs and ensure parents are accordingly informed.

## Figures and Tables

**Figure 1 children-12-01148-f001:**
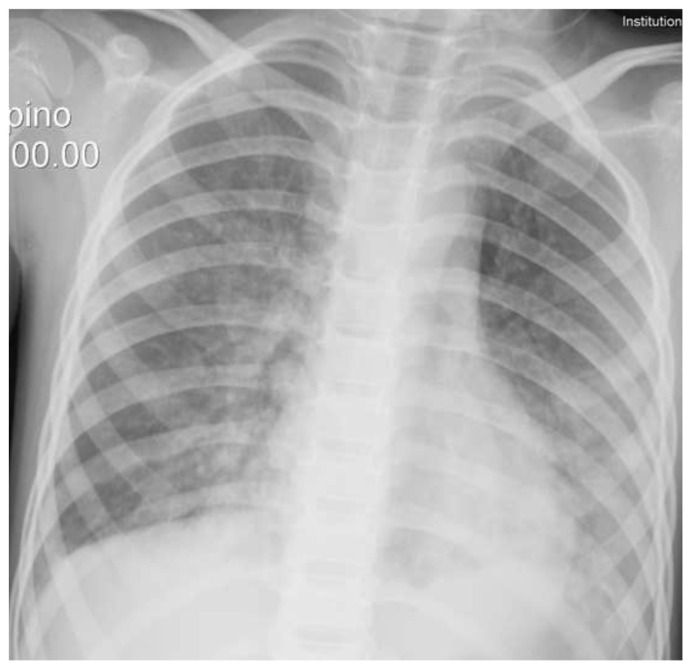
Chest X-ray showing bilateral lung infiltration, with a consolidation area over the right lung.

**Figure 2 children-12-01148-f002:**
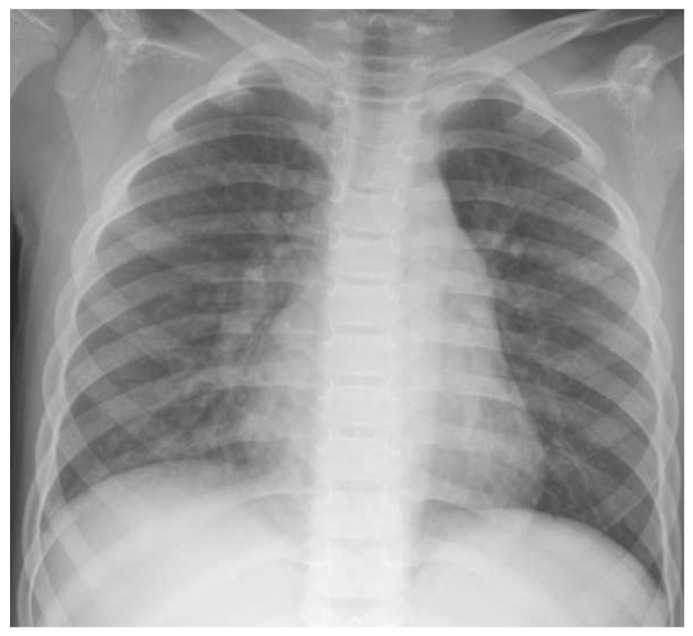
Chest X-ray showing bilateral thickening of the lungs’ interstitial tissue.

**Figure 3 children-12-01148-f003:**
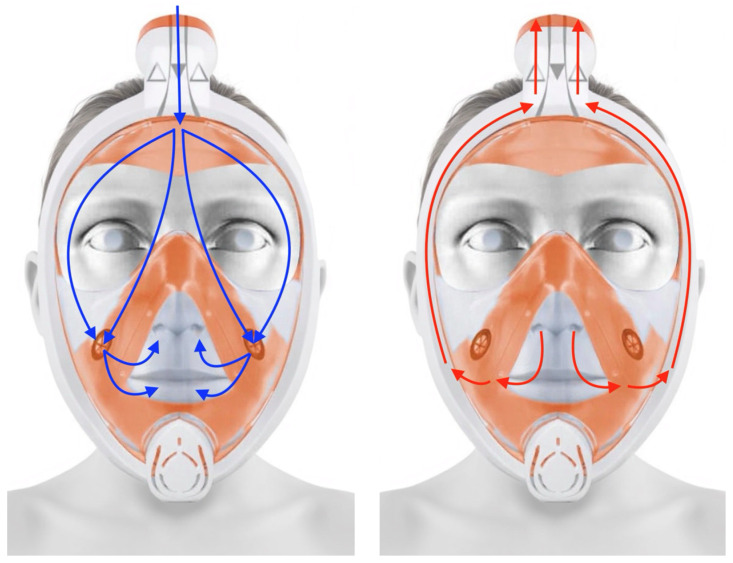
On the left side of the image, the inspiratory airflow is shown in blue. On the right side of the image, the expiratory airflow is shown in red. Modified from Grundemann et al. [3], 2023.

**Table 1 children-12-01148-t001:** This table shows the main potential problems associated with the mask.

Component	Intended Function	Potential Problem/Risk
**Valves**	Ensure unidirectional airflow and internal compartmentalization.	May stick open/closed or become obstructed (sand, salt, or debris) → risk of CO_2_ rebreathing and hypoxia.
**Seals**	Separate inhalation and exhalation pathways.	Loss of integrity (defects or poor fit) → leakage, rebreathing, and visor fogging.
**Silicone material**	Maintain seal integrity.	Degradation from heat or poor manufacturing in copies → risk of rebreathing.
**Float system**	Prevent water from entering snorkel.	Child may lack strength to clear it after closure → risk of impaired airflow.
**Head straps**	Secure mask to face.	(1) Difficult removal in emergencies. (2) If loose, water ingress → aspiration risk.

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
