# Peer review of "Full-Face Snorkeling Masks Carry a Risk of Hypercapnia and Drowning in Younger Children: A Case Series"

_children, 2025, doi:10.3390/children12091148_

Round 1
Reviewer 1 Report
Comments and Suggestions for Authors
In my view this is a valuable report highlighting the pitfalls of using full face snorkel masks (FFSM), particularly in children. As pointed out by the authors, these masks have associated problems, which include hypercapnia, hypoxia and are difficult to remove (as reported by the authors), but also introduce elevated breathing resistance and the potential for immersion pulmonary oedema.
Some suggestions:
- Abstact, 4th line: ... children due to their "effective marketing and relative" comfort ...
- 5th line: .. that, in case of malfunctioning or if used by young children, may trigger a mechanism of suggest replace with: "that may readily cause CO2 rebreathing, especially in young children".
- Abstract - Case presentation: the term "near drowning" is no longer encourage and is replaced by "non-fatal drowning". Suggest you rephrase to: ... department following non-fatal drowning. All incidents ....
- Abstract - Conclusions: Suggest change to: "particularly in younger children"
- Introduction- P2, para 1: Suggest you first cite Grundermann et al here.
- Introduction- P2, para 1: Suggest: “… organizations such as Duke University have initiated …” DAN was not mentioned in this Duke study.
- Introduction- P2, para 1: Suggest “A study of the safety ….. and replace reference with the subsequent paper – Farrel J, Natoli MJ …. Testing of full face snorkel masks to examine recreational snorkeler deaths. Undersea Hyperb Med. 2012; 49(1):515-528.
- Introduction- P2, para 2: replace “near” with “non-fatal”.
- Discussion – P7, para 2. Question: Is the dead space in the child’s device respiratory pocket the same as in the adult’s (i.e., 250 ml)?
- Discussion – P7, para 8, line 2. Suggest replace “shortly” with “quickly”.
- Conclusions – para 1. You shouldn’t introduce references in the Conclusions. If you want to cite this you should do it earlier in the paper.
- Conclusions – para 2. Suggest: “… available literature, reinforces concerns about the safety of FFSMs, especially in children under the age of six, and poses ….”
Author Response
Dear reviewer, thank you for taking the time to read our work and leave thoughtful comments.
You can find our responses to the points you raised attached.
Best regards.

Reviewer 2 Report
Comments and Suggestions for Authors
Thank you for the opportunity to review this manuscript. It is an interesting case report about an every day issue touching respiratory health.
The introduction is good. The cases are well presented, although an investigation of the individual FFSM would be nice to have to clarify valve dysfunction as a potential mechanism, e.g. aspiration, CO2 retention etc.
The discussion is far too long and should be significantly shortened e.g. by reducing doublings and incorporation of tables (e.g. causes of malfunction and potential consequences etc.). The issue of aspiration has not been mentioned.
Furthermore, the potential mechanism of injury due to the FFSM should be individually described in every case presented.
Author Response

(The authors gave the same response as above.)

Reviewer 3 Report
Comments and Suggestions for Authors
The topic was original and relevant: I consider it relevant particularly as important preventative information for people, and I mean those who should be supposed experts in children health care, that is pediatricians and GPs, but are not versed nor informed about diving related issues and risks and may be superficially thinking: "what real risks there can be by just snorkeling at the water surface and enjoy the view"?
The topic addressed a specific gap in the field or adds to the subject area: actually not very much, the matter of FFSMs safety is amply known and debated, but this paper raised an issue and flagged the special case of their use in very young children, which was not so specifically covered in other similar papers.
Based on logical reasoning and hypotheses, the argument and the conclusions convey a prudence message aimed at non diving medicine experts ( be it Physicians, Parents, or even Educators or Baby Sitters), who may consider a FFSM just an amusing and safe toy.
The manuscript is an interesting and useful read - and clinical reference- notwithstanding its relative simplicity
Author Response
Dear reviewer, thank you for taking the time and effort to read our work.
We truly appreciate the kind words highlighting the value of our study, even in its simplicity.
Best regards.
Round 2
Reviewer 2 Report
Comments and Suggestions for Authors
Thank you for the revision of the manuscript. The additional information are fine for me